# Zoo Animal Welfare Assessment: Where Do We Stand?

**DOI:** 10.3390/ani13121966

**Published:** 2023-06-12

**Authors:** Oriol Tallo-Parra, Marina Salas, Xavier Manteca

**Affiliations:** 1School of Veterinary Science, Universitat Autònoma de Barcelona, Campus UAB, 08193 Barcelona, Spain; oriol.tallo@uab.cat; 2Animal Welfare Education Centre, AWEC Advisors SL, Universitat Autònoma de Barcelona, Campus UAB, 08193 Barcelona, Spain; 3Antwerp Zoo Centre for Research and Conservation, Royal Zoological Society of Antwerp, Koningin Astridplein 20-26, 2018 Antwerpen, Belgium; marina.salas@kmda.org

**Keywords:** animal-based indicators, animal behaviour, stress, welfare assessment, welfare indicator, zoo animal welfare

## Abstract

**Simple Summary:**

Zoos and aquariums have recognized the paramount importance of animal welfare. Achieving and maintaining the highest possible standards of animal welfare is not only a moral obligation but also a necessary condition for zoos and aquariums to fulfil their educational and conservational functions. However, evaluating the welfare of zoo animals presents a challenge due to the wide range of species involved and a lack of knowledge regarding the specific requirements of each species. Therefore, to protect and promote the welfare of zoo animals, there is a need for science-based tools to assess and monitor it. This review article aims to discuss the advantages and limitations of existing methodologies for assessing animal welfare in zoos, describe the main animal-based welfare indicators for zoo animals, and identify areas that require further research.

**Abstract:**

Zoological institutions, such as zoos and aquariums, have made animal welfare a top priority, as it is not only a moral obligation but also crucial for fulfilling their roles in education and conservation. There is a need for science-based tools to assess and monitor animal welfare in these settings. However, assessing the welfare of zoo animals is challenging due to the diversity of species and lack of knowledge on their specific needs. This review aims to discuss the advantages and disadvantages of existing methodologies for assessing zoo animal welfare through: (1) A critical analysis of the main approaches to zoo animal welfare assessment; (2) A description of the most relevant animal-based welfare indicators for zoo animals with a particular focus on behavioural and physiological indicators; and (3) An identification of areas that require further research.

## 1. Introduction

Animal welfare has become an absolute priority for zoos and aquaria (from now on, zoos). Ensuring the highest possible standards of animal welfare is not only a moral duty but also a necessary condition if zoos are to realize their educational and conservational functions. Having science-based tools to assess animal welfare is needed to identify welfare problems and to monitor progress when improvement strategies are implemented. However, assessing the welfare of zoo animals is challenging due to, among other reasons, the sheer diversity of species kept in zoos and the lack of knowledge on the general biology and specific needs of many of them.

The objectives of this review are: (1) To discuss the main advantages and disadvantages of the existing methodologies to assess zoo animal welfare; (2) To describe the main animal-based welfare indicators for zoo animals and discuss their limitations; and (3) To identify areas in the field of zoo animal welfare that require further research.

## 2. Fundamental Principles of Zoo Animal Welfare Assessment

Methodologies to assess zoo animal welfare must be based on our current understanding of the concept of animal welfare. Historically, animal welfare has been defined using different approaches (see Fraser et al. [1] for a review), which can be grouped into three categories: biological functioning, emotional state, and “naturalness”. Each of these approaches has its own merits but none of them captures on its own the different aspects of animal welfare. It has been suggested, therefore, that the assessment of animal welfare must include all three approaches [1]. In fact, it is now widely accepted that animal welfare encompasses not only the physical health of the animals (i.e., the absence of diseases and injuries) but also their behaviour and emotions [2,3]. Behaviour is an essential element of welfare, among other things, because the possibility to engage in highly motivated behaviours contributes to animals experiencing positive emotions ([4], see below). In summary, when assessing zoo animal welfare, it is important to remember that the concept of animal welfare is broader than that of physical health, understood as the absence of diseases and injuries, since animal welfare includes both the physical and the mental state of the animals. Therefore, any assessment methodology must include both aspects.

For many years, the Five Freedoms and provisions [5] have offered a useful framework for the identification of welfare problems in animals. The Five Freedoms were initially developed for farm animals but have also been used for zoo animals. In recent years, the Five Freedoms have been criticized mainly because they fail to capture our current understanding of the biological processes underlying animal welfare [6] and because they are limited almost exclusively to the absence of negative situations, paying less attention to the importance of the positive aspects of well-being. Over the last few years, several authors have emphasized the so-called “positive welfare”, that is, the fact that to reach true animal welfare, it is not enough to guarantee the absence of suffering in animals, but that we must also provide them with the necessary conditions for them to experience positive emotions [4].

As an alternative to the Five Freedoms, the so-called Five Domains Model for assessing animal welfare was developed to address these problems. According to this model, the welfare of an animal results from its global emotional state, that is, from the balance between the positive and negative emotions that the animal experiences at a certain moment or over a period of time. This global emotional state constitutes the fifth domain of the model. The model recognizes four other domains, which are the “physical” domains (feeding, environment, health, and behaviour) from which positive and negative emotions are derived, which, when combined, define the fifth domain [7,8].

## 3. Main Approaches to Zoo Animal Welfare Assessment

There are several approaches to zoo animal welfare assessment, and they can be broadly grouped into five main categories:Species-specific protocols;Generic protocols and risk assessment methods;Assessment of welfare based on time budgets;Keepers’ ratings;Cognitive bias testing.

These approaches are not mutually exclusive, and, for example, some zoos combine keepers’ ratings with measurements of other animal-based and environment-based welfare indicators [9].

A brief description of each of these approaches, as well as a discussion of their advantages and disadvantages will follow.

### 3.1. Species-Specific Protocols

For a few species of zoo-kept animals, there are species-specific welfare assessment protocols that include a set of indicators and a description of the methodology to measure them. Some of these protocols are based on the Welfare Quality© protocols that were initially developed for farm animals kept under intensive production systems [10]. The Welfare Quality© protocols include four animal welfare principles (feeding, environment, health, and behaviour), which coincide with the four physical domains of the Five Domains Model. In turn, each principle includes several animal welfare criteria and, finally, each criterion is evaluated through one or several indicators. The welfare principles and the welfare criteria are the same regardless of the species (see Table 1), whereas the welfare indicators can vary across species.

One of the advantages of the Welfare Quality© protocols is that they combine animal- based indicators (such as behaviour, clinical signs, and body condition, among many others) and resource-based indicators (such as space available to the animals, for example). Traditionally, animal welfare used to be assessed through resource-based indicators, mainly because they are easier to measure and require less time and training. However, resource-based indicators sometimes fail to provide accurate information on the welfare state of animals and animal-based indicators are often preferred [11]. This is because the effect of a given environmental feature on the welfare of animals can vary across individual animals. Additionally, environmental features often interact with each other, and their effect on the animals may be difficult to predict.

Based on the Welfare Quality© assessment protocols, Clegg et al. [12] developed a welfare assessment protocol for bottlenose dolphins (*Tursiops truncatus*) under human care. The protocol included a total of 36 measures (more than half of which were animal-based) that were tested for feasibility and accuracy. Although the protocol has some limitations (including its restricted applicability to very young and very old dolphins), it is a very useful step towards a standardised welfare assessment tool for dolphins under human care, and has led to the development of other welfare assessment protocols for this species [13].

Similarly, Salas et al. [14] adapted the Welfare Quality© assessment protocol for cattle to measure the welfare of captive dorcas gazelles (*Gazella dorcas*). The suggested protocol includes 23 indicators and was proven to be useful in identifying welfare improvement opportunities in the five groups of dorcas gazelles at the three different zoos where the protocol was tested.

A slightly different approach was followed by Yon et al. [15], who developed a behavioural protocol to assess the welfare of captive African and Asian elephants (*Loxodonta africana* and *Elephas maximus*). The protocol includes three sections: Qualitative Behavioural Assessment (QBA), daytime behaviour questions and nighttime observations. QBA is a methodology that was initially developed for farm animals to assess the valence of an animal’s emotional state by measuring its demeanor ([16], see below). The items included in the protocol were tested for feasibility, validity, and reliability. The authors suggested that the protocol could be used together with other elephant welfare assessment tools that focus on health and physical condition.

Species-specific protocols have, at least in theory, several advantages over other welfare assessment methods, as they are meant to cover all aspects of animal welfare, use measures that have the potential of being tested for validity and reliability, and are tailored to the biological needs and peculiarities of each species. However, their main limitation is that very few of them have been developed until now. In fact, for the vast majority of zoo-kept species, there is a lack of validated indicators that can be integrated into a protocol. For example, despite being very charismatic species that have probably attracted more research efforts than many others, the number of fully validated welfare indicators for polar bears (*Ursus maritimus*) and African and Asian elephants is very small (see Skovlund et al. [17] for a review of welfare indicators in polars bears and Williams et al. [18] for a review of welfare indicators in elephants).

### 3.2. Generic Protocols and Risk Assessment Methods

Due to, in part, the lack of species-specific protocols for the vast majority of zoo-kept animals, several authors have proposed generic assessment protocols, i.e., protocols that can be used in any species. One example of these generic protocols is the welfare assessment tool proposed by Brando and Buchanan-Smith [19] as part of their 24/7 welfare framework. The authors adapted and expanded the 12 criteria of the Welfare Quality© assessment protocols, adding two more criteria. The resulting welfare tool is meant to be used by zoo staff to find out if their animals’ welfare needs are met.

Sherwen et al. [20] developed a welfare risk assessment protocol that includes a total of 20 indicators (both animal- and resource-based), as well as an scoring methodology, and each indicator was given a value of 0, 1 or 2. This method is meant to identify potential welfare issues and prioritize improvement actions so that zoo personnel can take a proactive approach rather than simply flag welfare problems when they have already appeared.

Generic protocols and risk assessment methods are obviously more flexible than species-specific protocols, as they are designed to be applied to any species. A main limitation, however, is that they can only be successfully applied if the biology and welfare requirements of each species are well-known, which is not always the case. Recently, new proposals have been suggested to, at least partially, overcome some of these limitations [21].

### 3.3. Assessment of Welfare Based on Time Budgets

Welfare assessments based on time budgets follow the assumption that the proportion of time that an animal spends in positive and negative behavioural states reflects its overall welfare.

This methodology was developed by Watters et al. [22], who provide a list of positive and negative behaviours, and suggest a methodology to obtain the above-mentioned ratio. One of the main advantages of this approach is that it gives an overall score of welfare that allows zoo personnel to follow possible changes in the welfare of their animals. Additionally, although the method requires behavioural observations, it is likely to be less time-consuming than some species-specific protocols, for example.

One of the main limitations of time budgets as a basis for welfare assessment is that it is not always easy to decide if a given behaviour is positive or negative. For example, an animal can be inactive because it is resting and content, or because it is bored. Additionally, time spent engaging in a given behaviour may not be enough to assess its welfare relevance. For example, aggressive interactions can have very negative effects on animal welfare even if they are very brief. Finally, time budgets provide information on the behavioural aspects of welfare, but not on the health or physical state of the animals.

Time budgets can be used to work out behavioural diversity, which has been proposed as an indicator of positive welfare. The rationale for using behavioural diversity as an indicator of good welfare is that a high behavioural diversity indicates that the behavioural needs of the animals are being met, as animals can display a wide repertoire of natural behaviours. On the contrary, when animals are unable to show their natural behaviours and engage in repetitive behaviours or become lethargic (which would be indicative of poor welfare), behavioural diversity will decrease [23]. There are several methods to calculate behavioural diversity and the so-called Shannon’s diversity index is the method that is most frequently used [24]. The validity of behavioural diversity as an indicator of positive welfare has been criticised based on both methodological problems and because the calculation of the diversity index does not consider the valence of the behaviours, i.e., whether a given behaviour reflects a positive or a negative welfare state [25]. These criticisms should stimulate further research on the validity of behavioural diversity across different taxa.

### 3.4. Keepers’ Ratings

Keepers can provide useful information on the welfare of their animals by observing their overall behaviour and demeanour. Several methods can be used to systematically gather keepers’ observations. One of these methods is the already mentioned ‘Qualitative Behavioural Assessment’ or QBA. This holistic assessment of welfare is based on the subjective impressions of observers and was initially developed for farm animals. QBA is now an important element in many welfare assessment protocols used in domestic species, including the Welfare Quality© assessment protocols described earlier. To carry out a QBA, several observers rate the animals on a number of descriptors, such as ‘happy’, ‘bored’, ‘relaxed’, ‘tense’, etc. The results of the QBA have a high inter-observer reliability and are correlated with quantitative measures of welfare in farm animals [16].

A similar approach can be used in zoo animals, and zookeepers’ qualitative assessment of individual animals’ welfare can be integrated into the daily routines of zoos [26]. It has been shown that zookeeper ratings also have a high inter-observer-reliability and are correlated with quantitative measures of welfare, both physiological and behavioural. For example, clouded leopards (*Neofelis nebulosa*) that were highly rated by their keepers as “tense” and “pacing” were found to have higher overall faecal glucocorticoid metabolite concentrations—an indication of stress and arousal—than leopards whose keepers did not rate them highly for these traits [27].

One of the main advantages of keepers’ ratings is that they require less time than other assessment methods, particularly those that are based on systematic observations of behaviour. This advantage can be extremely important, as many zoos face staff and budgetary constraints [26]. However, one problem with this methodology is the possibility that preconceived and untested ideas about the needs of animals can bias the welfare ratings provided by keepers. Another possible limitation of making a qualitative assessment of welfare based on the animal’s demeanour is that it can be more difficult for non-mammalian species.

### 3.5. Cognitive Bias

The term “cognitive bias” describes the changes in cognitive function (i.e., information processing) caused by the emotional state of the animal. Cognitive bias testing (and, more specifically, judgement bias paradigm) has emerged as the most promising method of assessing an animal’ emotional valence. In this paradigm, animals are trained to recognize one cue as predicting a positive event and another cue predicting a less positive/negative event and are then presented with ambiguous (intermediate) cues. The hypothesis—supported by many studies in a variety of species—supports the idea that animals in a negative affective state will be more likely to respond to these ambiguous cues as if they predict the negative event than animals in a more positive state [28].

Judgement bias testing has also been used in wild animals under human care, although very few studies have been conducted in zoo settings [29]. Nevertheless, this methodology appears to be promising in improving the welfare of zoo animals; perhaps not so much as a direct measure of welfare that can be used in regular welfare assessments, but as a method of validating other measures that can then be used more easily [29].

A summary of the advantages and disadvantages of the approaches to zoo animal welfare discussed in this paper can be found in Table 2.

## 4. Animal-Based Welfare Indicators for Zoo Animals

In general, welfare indicators are divided into two main categories: environment-based indicators (also known as ‘input-based or resource-based indicators’) and animal-based indicators (also known as ‘output-based indicators’) [30]. Among numerous others, the size and layout of the facilities, ambient temperature, and type and quantity of food are considered environment-based indicators. While environment-based indicators are widely used and known for their reliability and feasibility, this review does not cover these indicators. Conversely, the main animal-based indicators for animal welfare are behaviour, body condition, existence of clinical signs, fur or feather appearance, as well as diverse physiological and biochemical parameters.

Behavioural indicators are the most commonly used indicators for evaluating animal welfare [31]. This may be because behavioural observations are considered a non-invasive method of data collection [32], for which advanced technology is usually not required [31] and can be relatively inexpensive to apply [33].

Each individual is inherently unique, leading to diverse experiences and distinct needs. Consequently, these individual differences can significantly influence animal welfare. Specifically, these differences can be best characterized as variations in personality, reflecting stable behavioural differences observed across time and situations [34] (p. 654). Personality traits are influenced by a combination of genetic and environmental factors, resulting in the widely acknowledged notion that individual animals possess distinct personalities that can impact their welfare when kept in captivity [35,36]. Moreover, zoo animals have a wide range of behaviours in their repertoire, and therefore, may respond differently to conditions that could potentially impact their welfare [33]. Consequently, it is not possible to rely on a single behaviour-based indicator to assess an individual’s welfare. The most commonly used welfare indicators based on behaviour are those that assess abnormal behaviours and changes in the expression (frequency, duration, or intensity) of normal behaviours [30].

### 4.1. Indicators Related to Abnormal Behaviours

For this revision, we will consider the behaviours observed in captivity that are rarely or never seen in the wild [37] and that are a consequence of poor welfare as abnormal. However, this does not necessary imply that all abnormal behaviours indicate a welfare problem.

#### 4.1.1. Abnormal Repetitive Behaviours

Abnormal repetitive behaviours (ARB), also called stereotypies, were initially defined as repetitive, invariant behaviours without an apparent immediate function [38]. An updated definition considers stereotypies to be repetitive behaviours caused by frustration, repeated attempts to adapt to the environment and/or a dysfunction of the central nervous system [39]. This updated definition takes into account that these behaviours may not be as invariant as previously assumed, and that some of them might actually aid animals in adapting to challenging or unsuitable environments [30,39].

In zoo animals, ARB can manifest in various ways, and the frequency of each type may differ depending on the taxonomic group [40]. For example, locomotor repetitive behaviours (such as pacing) are more commonly observed in carnivores, where the animal moves repetitively along the same path. Oral repetitive behaviours (such as licking or biting objects) are more frequently observed in ungulates. Primates tend to show ARB related to repetitive body movement without displacement. Oral and locomotor repetitive behaviours have also been described in birds. There is still little information on possible abnormal behaviours developed in reptiles, amphibians, and fish [32]. However, in some reptile species [41], individuals have been observed to repetitively interact with transparent barriers (such as glass), and in fish, possible ARB including repetitive swimming patterns have been observed [42].

Repetitive behaviours are probably the abnormal behaviours that cause the most concern in zoos because they are very obvious when present and can cause anxiety to visitors when they observe them [32,43]. They are also some of the most used welfare indicators and are very useful, as they can indicate welfare problems related to frustration, stress, behavioural restrictions [39] and the onset of medical problems [30]. However, the use of ARB as welfare indicators can be confusing because the relationship between an ARB and welfare can sometimes be complex. For example, an ARB that has existed for a long period of time may become ‘fixed’, and therefore, there is a possibility that an animal currently maintained in an appropriate environment shows an ARB as a result of a previous unsuitable environment [44].

#### 4.1.2. Damaging Behaviours

##### Self-Injurious Behaviours

Self-injurious behaviours (SIB) have been linked to poor welfare in captivity. In captive birds, particularly psittacines, feather damaging behaviour (FDB) has been reported as a behaviour where birds pluck their own feathers [45], which is considered abnormal and associated with the inability to perform natural behaviours such as foraging and podomanipulation [46,47].

Captive primates have been reported to exhibit hair-plucking behaviour [37], which resembles FDB, as well as other SIB such as self-biting and head-banging [48]. These abnormal behaviours present a significant and immediate threat to an individual’s physical health and overall welfare [49].

##### Regurgitation and Reingestion

Another abnormal behaviour observed in captive primates is the intentional regurgitation and reingestion of previously ingested material [50,51,52], which has also been observed in marine mammals. This behaviour can lead to health problems and is believed to be an adaptive response to boredom, stress, space restriction, lack of control over the environment, dieting, or the inability to develop normal foraging behaviour [53].

#### 4.1.3. Apathy

The abnormal state of inactivity and lack of response to environmental stimuli in animals is referred to as apathy [30]. In humans, this condition is commonly linked to depression [54,55]. Animals may exhibit apathy in monotonous or stressful environments, where they feel they have no control, and when they are experiencing pain [56].

However, it is essential to note that there are individual and species differences when assessing apathy in zoo animals. To distinguish between normal levels of inactivity and apathy-associated inactivity, one must consider the typical activity rhythm of each species, including whether they are nocturnal or diurnal animals [30].

### 4.2. Indicators Related to Changes in the Expression of Normal Behaviours

A good understanding of the species being assessed (including individual differences) is necessary to identify a normal range of activity or deviations from it. Unfortunately, such knowledge is lacking for many zoo-housed species, as their needs in natural habitats are not always fully understood [33].

When using normal behaviours to assess welfare, we should pay attention to changes in their expression, specifically changes in the frequency, intensity, and/or duration of these behaviours [30]. Therefore, it is important to regularly measure the normal behaviour of animals and compare it to detect changes in the expression of these behaviours.

#### 4.2.1. Social Behaviours

##### Affiliative and Agonistic Behaviours

Social interactions, including both affiliative (or positive) and agonistic (or negative) behaviours, are a normal part of the behaviour repertoire in all species. However, an excess or high intensity of negative behaviours may indicate a welfare problem and can cause injury and stress. Negative emotions such as pain, fear, chronic stress, and frustration can cause or increase aggressive behaviour [30]. Conversely, affiliative behaviours are generally considered rewarding and can have a buffering effect on stress and reduce social tension, improving group cohesion and bonding between individuals. Therefore, affiliative behaviours are often considered indicators of positive welfare [57]. However, an increase in affiliative behaviours does not always indicate good welfare. For example, after a conflict between two or more animals in a group, an increase in affiliative interactions may occur, a phenomenon known as ‘behavioural reconciliation’ [58].

##### Maternal Behaviour

Maternal behaviour is another social behaviour that can be affected by stressful situations, pain, or other negative emotions. Negative emotions can lead to abnormal maternal behaviours such as neglect or aggression towards the young, excessive grooming of the young, or frequent movements of them from one place to another [30].

##### Play Behaviour

Play, whether it happens with one’s self or socially, is an enjoyable activity that usually only takes place when other needs are fulfilled. In fact, the existence of play behaviour might suggest that the animal is not deprived of significant sources of pleasure and that other biological needs are satisfied. Therefore, play behaviour is generally viewed as a positive indicator of animal welfare, particularly in mammals and birds [57]. However, caution should be exercised when using play behaviour as a positive indicator, as social play in some species can sometimes lead to aggression. Furthermore, there are cases where it was observed that the occurrence of play behaviour can increase following a period of stress or deprivation of opportunities to play [30].

#### 4.2.2. Maintenance Behaviours

##### Food Intake

If management practices are not improved, food in zoos is typically presented in a simple and direct way, such as in a feeding bowl or at a single distribution point. As a result, foraging and consummatory behaviours, such as chewing, take significantly less time than they would in the wild, particularly if food availability is limited. This can lead to animals having unsatisfied motivations to perform these natural feeding activities [59].

However, a reduction in food intake may also result from an intense stress response, which can be influenced by factors such as the type and intensity of the stressor. It should be noted, though, that stress can occasionally increase feed intake, potentially leading to obesity and other health problems [30].

##### Rumination

Rumination is a specific feeding function, essential for optimal digestion in ruminants. The duration of rumination is affected by the type of diet, and insufficient fibre intake leads to a decrease in rumination time, which can result in gastrointestinal problems [59]. Furthermore, stress can decrease rumination time [60], and inadequate opportunities for rumination may lead to the development of oral repetitive behaviours [61].

##### Sleep Behaviour

Sleep behaviour or sleeping is defined as a motionless state in which an animal is not alert and has both eyes closed if anatomically possible. Stress can interrupt sleep patterns, reducing the duration and quality of sleep [62]. For this reason, sleep behaviour can be an important welfare indicator, although not easily observable at night without the use of technology (e.g., infrared cameras).

#### 4.2.3. Behaviours Related with Exploration and Interaction with the Environment

##### Anticipatory Behaviour

Anticipatory behaviour is a set of behaviours expressed by animals before acquiring a positive outcome or resource [63]. Some examples of the anticipated positives outcomes are opportunities for reproduction, positive social interactions, or food, as well as behavioural opportunities to obtain primary reinforcements (e.g., positive reinforcement training or environmental enrichment). Anticipatory behaviours occur in an area close to where the positive event takes place, their intensity or frequency increases as the time of a predictable positive outcome approaches, and stops being expressed when the motivation is consummated [64]. Anticipatory behaviours themselves are not indicators of positive or negative welfare, but rather the intensity with which they are expressed is related to animal welfare. For instance, the intensity of anticipatory behaviour tends to decrease when animals have more opportunities to obtain rewards, and conversely, increases when opportunities are scarce.

##### Use of Enclosure

Studying how animals use the enclosure in which they live can help quantify the effects of environmental enrichment, modifications or improvements in facilities, and changes in social groups [65]. It can also help detect each individual’s preferred areas and locate where the most valuable resources for the animals are. This can allow zoos to make informed, evidence-based management decisions and redesign areas avoided or underutilized by animals to maximize the enclosure’s potential.

#### 4.2.4. Other Behaviours

##### Displacement Behaviours

Displacement behaviours are behaviours that appear irrelevant and inappropriate in the context in which they appear [66]. Some self-directed behaviours (actions directed at an animal’s own body) are considered displacement behaviours resulting from frustration and/or a situation of internal conflict in an animal. They are often linked to negative excitement and even used as indicators of stress or welfare problems in primates [67]. However, several studies indicate that not all displacement behaviours have the same function, and some of them, such as self-scratching, can also increase with positive excitement [68]. For this reason, and because these may be normal behaviours that are unrelated to the animal’s emotional state, it is necessary to have long-term records of the frequency of displacement behaviours if they are to be used as indicators of welfare [30].

##### Vocalisations

Animal vocalizations are sounds that animals actively produce and can reflect their emotional states. The type of vocalization may differ among individuals based on their personality and among species. For instance, in highly emotional circumstances, such as when confronted with predators, some animals may produce alarm calls frequently, while others may produce them less often or remain silent [69]. Nevertheless, combining vocalizations with other indicators of emotions can provide a tool for monitoring both positive and negative emotions, which in turn can be used to assess animal welfare [70].

### 4.3. Physiological Indicators of Welfare: Using Physiological, Pathophysiological, Cellular and Biochemical Biomarkers to Assess Welfare

Any physiologically, pathophysiologically, cellularly and/or biochemically measurable change that occurs when the welfare of an animal varies could be a potential welfare indicator. Because all these biological processes are related, the concept of “physiological indicators” is usually used to encompass and represent a large and highly diverse group of neuroendocrine, haematological, cardiovascular, immunological, and/or cellular welfare-related biomarkers, among others [71]. Typically, physiological indicators are separated into: (A) Indicators related to stress, emotions, and other adaptive states; and (B) Indicators of health or related to pathological processes. However, this separation is more practical than real since many biomarkers are modulated or modulate both processes [72]. Despite assessing the health status of an animal being a critical part of its welfare monitoring [73], most health-related welfare indicators will not be included in this review due to the vastness of the zoo veterinary medicine field.

#### 4.3.1. Physiological Welfare Indicators: General Advantages and Limitations

As some central elements of animal welfare are neuroendocrinological processes, such as emotions or stress, some physiological indicators may offer a very interesting approach to these welfare elements through the measurement of biomarkers that are directly involved in these processes. Physiological indicators can provide information about aspects of the welfare of an animal that may be challenging to obtain using behavioural indicators [74]. This makes them particularly useful for certain species, individuals, or specific circumstances related to both the data sampling context and the welfare assessment goal. Physiological indicators are also valuable in welfare assessments because of their variety and ability to be used in combination with, and complementarily to, other welfare indicators such as behavioural indicators [30,75].

Generally, the use of physiological indicators is more expensive and involves more complex sampling than using behavioural indicators, especially in those individuals that are not trained for sample collection. However, physiological indicators also have advantages for use in zoo-housed animals. First, collecting physiological indicators requires less time spent in the facility than collecting behavioural ones. Second, some physiological indicators can be measured in matrices obtained without any contact (not even visual) with the animal (for example, analysing biomarkers in shed feathers [76]), or in matrices taken opportunistically in interventions that involve manipulation of the animal for other reasons. Third, some physiological indicators enable retrospective welfare assessments and can be used to evaluate welfare over different timeframes [77].

Importantly, researchers and other animal welfare evaluators that are not familiar with physiological indicators tend to perceive them as more objective and easy-to-interpret welfare indicators than other. This is misleading, as a critical limitation of physiological indicators is the need for a substantial understanding of the indicator and the species being studied, as well as the individual’s particularities and context, for correct interpretation [78,79]. Physiological indicators can vary or change differently, for example, by increasing, decreasing or not varying their concentrations at all, depending on the species assessed or the nature, duration or severity of the studied welfare problem [71,80,81]. Moreover, physiological indicators may be influenced by factors unrelated to the well-being of the animals, commonly referred to as confounding factors. These factors may include biological rhythms (circadian rhythms, seasonal changes, etc.), differences associated with sex or age, and the effects of other biological processes such as pregnancy or physical activity, among many others [82,83]. Knowing and understanding the confounding factors that may be affecting the physiological indicators used in each specific case is critical for their correct interpretation. Another limitation of physiological indicators is that most exhibit high inter-individual variability, which adds an extra difficulty (usually unexpected) to their interpretation [84,85]. Furthermore, when using physiological indicators that require laboratory work, it is essential that the sampling, storage, processing, and analysis procedures of the samples are correct and validated for each species, indicator and matrix [86,87].

Meeting the necessary requirements to correctly use physiological indicators is always challenging. While these indicators can provide valuable insights into the welfare of wild animals (both under human care and in their natural habitat), their use requires in-depth knowledge of the characteristics of each indicator and matrix, individual, species, and context [88,89] and appropriate sampling design and execution [90,91].

##### Selecting the Right Matrix Is as Important as Selecting the Right Indicator

Most physiological indicators are required to be extracted from a biological matrix and quantified, and there are many matrices that can be used for this purpose [77]. As each biological matrix has a unique composition, growth/renovation rate, and mechanisms for biomarker accumulation, the way a specific biomarker represents the physiological state and well-being of an animal can change completely depending on its matrix of origin [77,92]. Although each matrix is different, we suggest a three-level practical classification in Table 3.

Despite their capacity to represent a welfare-related physiological status at different timeframes, other aspects of matrices are also important and must be considered. Matrices can vary in their degree of invasiveness, sampling difficulty, or storage requirements, and, importantly, all of them have their own confounding factors that must be known and considered [86]. In addition, the usefulness and characteristics of some matrices are still under discussion as some of them have only recently been suggested for welfare assessments [101,102]. Finally, the choice of the matrix used for biomarker quantification should also be made carefully, considering the objectives of welfare assessment (e.g., short or long term), sampling possibilities, and published evidence on the matrix for the biomarker and species studied [103].

#### 4.3.2. Physiological Indicators Related to Stress

The physiological indicators related to the stress response are particularly interesting, and the most employed physiological indicators [89], for several reasons. First, because the stress response is present, at least, in all vertebrates [104]. Second, because, as stress is an unspecific response, stress-related indicators are versatile and capable to respond to several welfare issues [105]. Finally, because certain stress-related indicators may be both a cause and a consequence of welfare problems (as it happens with other indicators such as damaging behaviours). For example, glucocorticoids can be modulated as a consequence of stress-related welfare problems [106]; however, in specific circumstances, they may also jeopardize welfare by suppressing vital behaviours or decreasing the animal’s long-term immunocompetence [107].

Despite the mentioned advantages, an important and common limitation in most stress-related physiological indicators is that the stress response is related to the degree of arousal of an individual rather than the valence of the experienced emotions [108]. Thus, an increase in the stress response has been described in activities such as playing or mating, which are unrelated to welfare problems or, in some cases, even positive for it [30,75,109]. Consequently, these indicators require knowing the context of the studied animals when used [109].

##### Glucocorticoids

Assessing concentrations of glucocorticoid hormones (cortisol in most mammals and fish, and corticosterone in birds, reptiles, and amphibians) is the most widely used option for quantifying stress response, specifically by measuring the activity of the hypothalamic–pituitary–adrenal/interrenal axis [31,105]. Glucocorticoids can be detected in several matrices, and the extensive existent knowledge about them as welfare indicators make them, when interpreted correctly within a context, some of the primary options for assessing stress in zoo animals, including for less known species [79,89]. Despite this, glucocorticoids are far from being free of confounding factors, limitations, and criticism [110,111].

##### Heart and Respiratory Rates

Changes in heart rate or an increase in its variability have been described as good indicators of some short-term welfare problems, provided that the context surrounding the individual, especially their physical activity and metabolic rate, is known [33]. However, they often require direct contact and immobilization, which could interfere with the obtained results or make a correct interpretation difficult in non-trained animals [112]. An increase in respiratory rate has also been described as a similar welfare indicator, and although it is not always easy to measure, it has the advantage of usually being assessed visually [71]. Respiratory rate, measured through opercular movements, is a commonly used indicator in various fish species [113].

##### Heterophil-to-Lymphocyte and Neutrophil-to-Lymphocyte Ratios

The heterophil-to-lymphocyte ratio in birds and reptiles, and the neutrophil-to-lymphocyte ratio in mammals, amphibians and fish can be used as stress-related indicators due to their relationship with circulating glucocorticoids [114]. However, this relationship is complex and these indicators require knowledge of the species’ leucocyte response and the health status of the target individual to be correctly interpreted [115]. Nevertheless, the assessment of leucocyte profiles has been widely used as a stress indicator, particularly in birds.

##### Immunoglobulin A

Concentrations of immunoglobulin A (IgA), particularly its secretory form, are influenced by the stress response, and can supposedly provide information about the affective state of individuals [57]. Moreover, the possibility of collecting IgA non-invasively and in various matrices, and its ability to represent the immune status of the animal make IgA an interesting indicator [75]. Unfortunately, there are still important aspects to be studied regarding IgA, such as modulatory and confounding factors, as demonstrated by some published contradictory results [116].

##### Dehydroepiandrosterone, Dehydroepiandrosterone Sulfate and Their Ratios with Glucocorticoids

The dehydroepiandrosterone and its sulfated version (referred to collectively as DHEA(-S)) are neuroprotective androgens that act as antagonists to glucocorticoids [117]. Although changes in their concentrations have been described in association with various welfare issues (mainly stress and pathological processes [118]), the relationship between DHEA(-S) and glucocorticoids is complex and remains poorly understood [119]. Nonetheless, several studies suggest that the cortisol/DHEA(-S) ratio could be particularly interesting for evaluating chronic stress [118].

##### Acute Phase Proteins

Acute phase proteins are proteins that participate in the acute phase response of the immune system [120]. Although they have been mainly used as indicators of tissue damage and inflammation [121], they have also been described as potential stress indicators [122]. However, there are significant differences among species in relation to their acute phase proteins and functions, and very few have been studied and validated in wild species [123].

#### 4.3.3. Other Physiological Indicators of Welfare

There are other physiological indicators of welfare that are not so directly related to the stress response (although they might be influenced by it). While some of these indicators have been well-studied, such as body condition, others, such as oxytocin or those related to telomeres, require further research to better understand their use in the welfare assessment of zoo animals.

##### Telomere Length and Attrition

The length and attrition of telomeres are used to measure biological age [123]. However, negative and positive states of welfare influence these phenomena, accelerating or reducing (even reversing) it, respectively [124]. Because of that, telomere length and attrition have been proposed as welfare indicators capable of integrating both positive and negative experiences over an animal’s lifetime [125]. Despite their attractive profile, the biological particularities of the telomeres of each species must be known, and their characteristics and uses as welfare indicators need to be studied [125].

##### Oxytocin

Oxytocin, a hormone related to positive social interactions and positive affective states [126], has recently gained significant interest as a welfare indicator due to the scarcity of physiological indicators of positive welfare states [127]. However, its use is limited by the current lack of knowledge on the oxytocinergic system, its confounding factors, and the real utility of oxytocin in welfare evaluations [128].

##### Body Condition

Body condition is an excellent welfare indicator for zoo animals. In zoos, obesity has been described as an important and frequent welfare problem [129], often associated with a range welfare deficiencies of diverse nature (nutritional, environmental and behavioural, among other). Despite its supposed simplicity, body condition should be systematically measured using validated indices, taking into account natural variations in each species and context [130].

##### Life Expectancy, Mortality, and Prevalence and Incidence of Diseases

When assessing welfare at a population level, there are additional population-based welfare indicators that can be used. The sensitivity of life expectancy, mortality, and prevalence and incidence of diseases are low, especially in modern zoos with high welfare standards and/or when working with small populations. Moreover, they should be interpreted with caution [131]. However, when all necessary information is available, they are capable of providing integrated information on the welfare status of a population over a long period of time [132].

##### Other Potential Physiological Indicators of Welfare for Zoo Animals

There are numerous indicators with the potential to aid in the assessment of welfare in zoo animals, and there will likely be more in the future. However, their usefulness in these non-domesticated species is yet to be confirmed, partly due to their recent emergence as welfare indicators and/or because most studies have been conducted in domestic species. Examples of such potential indicators include chromogranin A or alpha-amylase [119], changes in body or eye colour in certain fish species [133], and markers of oxidative stress, such as “leukocyte coping capacity” [134]. When preparing a welfare assessment of a zoo animal or group, it is highly recommended to conduct a scientific literature search to find potentially useful indicators valid for the species and context under evaluation.

## 5. Areas Deserving Further Research

The last few years have seen a significant progress in the scientific foundations of zoo animal welfare assessment and their practical applications. Examples of this progress include—but are not limited to—the development of species-specific protocols (e.g., [12,14,15]), the identification and validation of welfare indicators [17,18], the applications of experimental paradigms that were initially developed to evaluate the welfare of farm and laboratory animals to zoo animals (e.g., [29]), and fruitful discussions on the possible methodological and conceptual pitfalls and limitations of some welfare assessment approaches (see for example, [135]).

Despite this considerable progress, there are several areas that deserve significant additional effort. First, most research conducted so far on zoo animal welfare assessment has focused on a few charismatic mammals, particularly apes, dolphins, elephants, and some large carnivores. This may have been due to, at least in part, the pressure of animal rights groups. However, the zoo community must not forget that our concern for animal welfare comes, first and foremost, from our moral duty toward animals as sentient beings, and most scientists agree that many non-mammalian species are sentient [136].

Moreover, the conservational role of zoos is likely to be extremely important in relation to less charismatic species as well. Amphibians are a good example, as a large proportion of species are endangered in the wild and zoos could play a critical role in preserving them [137]. However, research on amphibian welfare is very limited. A similar situation applies to fish, reptiles and even many birds and mammals.

Second, even for the few zoo species that have been the subject of applied welfare research, the number of valid, reliable, and feasible indicators is still very limited (see for example, Skovlund et al. [17]). Therefore, it is urgent to identify and validate further indicators that cover all aspects of animal welfare for most, if not all, zoo species. Given the widely acknowledged importance of positive welfare, indicators of good welfare should be included as well. The use of new technologies for animal monitoring is a field that is expanding [138], with the potential to be very beneficial for welfare assessments in zoos. If further research confirms their usefulness, new technologies will not only facilitate practical, exhaustive, and long-term animal monitoring, but also the development of new welfare indicators or their practical application.

As a final thought, dialogue between zoo animal welfare scientists and researchers studying the welfare of domestic and laboratory animals is extremely important. Throughout this paper, we have highlighted the fact that both welfare assessment frameworks and indicators, as well as experimental paradigms that were originally developed for farm or laboratory animals have been successfully applied to zoo animals, and this is likely to hold true in the future. It is also important to strengthen the interchange of information between scientists working with animals in the wild and those working with wild animals under human care. This is by no means meant to imply that the behaviour of free-ranging animals should be taken as a gold standard to assess the welfare of zoo animals (see [139] for a discussion of the methodological and conceptual problems of such an approach). However, a sound knowledge of the natural history of a given species is fundamentally important to formulate research questions about its requirements in captivity.

## Figures and Tables

**Table 1 animals-13-01966-t001:** The principles and criteria that are the basis of the Welfare Quality© assessment protocols.

Welfare Principles	Welfare Criteria
Good feeding	Absence of prolonged hunger
	Absence of prolonged thirst
Good housing	Comfort around resting
	Thermal comfort
	Ease of movement
Good health	Absence of injuries
	Absence of disease
	Absence of pain induced by management procedures
Appropriate behaviour	Expression of social behaviour
	Expression of other behaviours
	Good human-animal relationship
	Positive emotional state

**Table 2 animals-13-01966-t002:** Some advantages and disadvantages of the main approaches to zoo animal welfare assessment.

	Advantages	Disadvantages
Species-specific protocols	Use measures that can be tested for reliability and validity (Some of them) cover all aspects of welfare Consider the characteristic of each species	Can be very time consuming There are only a few published protocols
Generic protocols and risk assessment methods	More flexible than species-specific protocols, as they are meant to be used with any species Cover all aspects of welfare	To be properly used, a sound knowledge of the biology and welfare needs of each species is needed Indicators not necessarily validated or specified/described in the protocol
Time budgets	Provide an overall score of welfare with a moderate time investment	Focus only on behaviour and do not provide information on health or physical state It is not always obvious if a given behaviour can be considered positive or negative
Keepers’ ratings	Require less time than other methods Facilitate the engagement of zookeepers in the process of welfare assessment	Preconceived ideas on the animals’ needs may bias the ratings Can be more difficult to use in species that are far away from humans in evolutionary terms (e.g., fish, reptiles, amphibians, etc.)
Cognitive bias testing	Can provide information on the overall emotional state of animals Can be useful in validating other measures that can be easier to use as part of routine welfare assessments	Can be time consuming, as animals have to be trained Very few studies performed so far in zoos

**Table 3 animals-13-01966-t003:** A three-level practical classification of matrices for quantifying physiological indicators.

	Description	Examples
Single-point matrices	These matrices provide information about the well-being of an animal at a particular moment in time, which is generally very close to the sampling moment or the minutes before.	Blood [93], saliva [94], and cutaneous mucus [95].
Intermediate matrices	These matrices accumulate biomarkers over a medium period of time (from a few to several hours) and can represent the welfare state that an animal had several hours before sampling and for a longer timeframe. Their renewal/excretion rates are also intermediate and must be considered.	Faeces [96], and fat [97].
Accumulative matrices	These matrices accumulate biomarkers over long periods of time, providing integrated and retrospective information on an animal’s well-being in the long term (days or weeks). Their renewal/excretion rate is usually low.	Hair [98], feathers [99], and fish scales [100].

## Data Availability

No new data were created or analysed in this study.

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
