# Peer review of "Zoo Animal Welfare Assessment: Where Do We Stand?"

_animals, 2023, doi:10.3390/ani13121966_

Round 1
Reviewer 1 Report
A very important manuscript for all those involved in the problematics of animals maintenance in zoos.
The literature collected for the review is absolutely sufficient to conduct reflections on the subject.
I accept the manuscript for printing in its current form, but I recommend that you pay attention to typing errors and changing the font color of literature item 103 in verse 531.
More graphically interesting tables would certainly enrich the reception of the manuscript, but of course they do not affect its substantive value, so I will leave the decisions to the authors.
Author Response
A very important manuscript for all those involved in the problematics of animals maintenance in zoos. --> Thank you very much for your time and review.
The literature collected for the review is absolutely sufficient to conduct reflections on the subject. --> Thank you.
I accept the manuscript for printing in its current form, but I recommend that you pay attention to typing errors and changing the font color of literature item 103 in verse 531. --> Thank you. Errors have been solved.
More graphically interesting tables would certainly enrich the reception of the manuscript, but of course they do not affect its substantive value, so I will leave the decisions to the authors. --> We have considered creating a table summarizing the indicators, but after adding it, we honestly believe it does not add much value to the manuscript and we would like to not add extra tables.
Reviewer 2 Report
Review article that summarizes the advantages and limitations of existing methods to assess animal welfare. This is very needed and timely for the current debates and need in the field to assess or “take stock” of current tools and practices and to reflect critically on where gaps are present and where have we made strong progress.
The authors do a good job out laying out the current protocols applied in welfare research, and present a combination and some history for what has been done and how has it been assessed. I found the organization and breakdown of assessment types, behavioral and physiological methods etc very clear and laid out well for the reader.
One aspect or question that this raises is since this is a review, is the growing body of literature that links applications of applied behavior analysis into welfare assessment and addressing welfare behavior concerns. There is no mention of this in the paper, and may warrant some attention as it relates to Self-injurious behaviors or as a future area of research. Additionally, I would consider adding some of the literature on Abnormal Repetitive Behaviors in terms of how the past experiences can "carry over" as well into the current environment even if the current environment meets the animals needs.
Overall, this is a strong paper that very cleanly gives an overview of current practices and where we are, and introduces where the field may want to go for continuing to look at questions of assessing animal welfare in zoos and aquariums.
Author Response
Review article that summarizes the advantages and limitations of existing methods to assess animal welfare. This is very needed and timely for the current debates and need in the field to assess or “take stock” of current tools and practices and to reflect critically on where gaps are present and where have we made strong progress. --> Thank you very much for your time and review.
The authors do a good job out laying out the current protocols applied in welfare research, and present a combination and some history for what has been done and how has it been assessed. I found the organization and breakdown of assessment types, behavioral and physiological methods etc very clear and laid out well for the reader. --> Thank you.
One aspect or question that this raises is since this is a review, is the growing body of literature that links applications of applied behavior analysis into welfare assessment and addressing welfare behavior concerns. There is no mention of this in the paper, and may warrant some attention as it relates to Self-injurious behaviors or as a future area of research. Additionally, I would consider adding some of the literature on Abnormal Repetitive Behaviors in terms of how the past experiences can "carry over" as well into the current environment even if the current environment meets the animals needs. --> A mention about Behavioural Diversity Index as a potential welfare indicator has been added in section 3.3. In lines 288-291 there is a specific mention about the “carry over” effects that past experiences (and past poor welfare conditions) can have on the current behaviour of an animal.
Overall, this is a strong paper that very cleanly gives an overview of current practices and where we are, and introduces where the field may want to go for continuing to look at questions of assessing animal welfare in zoos and aquariums. --> Thank you, very much appreciated.
Reviewer 3 Report
This is a really useful and informative review of the wide range of approaches that could be used to assess zoo animal welfare and I want to thank the authors for making this accessible to a range of audiences, particularly zoo practitioners who are currently navigating this field and may lack scientific training.
Table 2 is a fantastic summary of approaches and Section 4.3 clearly explains the pros and cons of the different welfare measures commonly used in studies but in practice may be difficult to achieve.
I have very minor suggestions for the manuscript which I hope will clarify a couple of points that might be ambiguous.
Line 124 - is there a word missing at the end of the sentence? Perhaps "...other templates"?
Lines 202-205 - I suggest an additional sentence at the end of this paragraph to ensure readers take care when using glucocortoid metabolites as a measure of 'stress' as it can also be indicative of other emotional states such as heightened excitement. The authors explain this further on in the review (particularly in section 4.3) but a mention here would be a useful addition to avoid confusion.
Lines 248 and 635 - The use of the term 'wild animals' could be interpreted in different ways. Are the authors still referring to zoo animals or animals in-situ? Or wild-derived animals? Consider rephrasing this on both lines to avoid confusion.
Line 587 - typo "inflammation" written twice in English and Spanish.
The quality of the English language is excellent and only a couple of minor typos were detected.
Author Response
This is a really useful and informative review of the wide range of approaches that could be used to assess zoo animal welfare and I want to thank the authors for making this accessible to a range of audiences, particularly zoo practitioners who are currently navigating this field and may lack scientific training. --> Thank you very much for your time and review.
Table 2 is a fantastic summary of approaches and Section 4.3 clearly explains the pros and cons of the different welfare measures commonly used in studies but in practice may be difficult to achieve. --> Thank you.
I have very minor suggestions for the manuscript which I hope will clarify a couple of points that might be ambiguous.
Line 124 - is there a word missing at the end of the sentence? Perhaps "...other templates"? --> Thank you. Clarified. The Clegg and colleagues protocol led to the development of other welfare assessment protocols for bottlenose dolphins.
Lines 202-205 - I suggest an additional sentence at the end of this paragraph to ensure readers take care when using glucocortoid metabolites as a measure of 'stress' as it can also be indicative of other emotional states such as heightened excitement. The authors explain this further on in the review (particularly in section 4.3) but a mention here would be a useful addition to avoid confusion. --> “and arousal” as been added: -an indicator of stress and arousal-
Lines 248 and 635 - The use of the term 'wild animals' could be interpreted in different ways. Are the authors still referring to zoo animals or animals in-situ? Or wild-derived animals? Consider rephrasing this on both lines to avoid confusion. --> Modified to avoid confusions.
Line 587 - typo "inflammation" written twice in English and Spanish. --> “inflamación” deleted.
Author Response
The introduction does a good job of reviewing the existing literature but needs more references on other points, as I suggest down. Furthermore is necessary explain more the novelty of the current research. --> Thank you very much for your time and review.
Comments on 1. Assessment of welfare based on time-budgets.
Maybe this need more consideration. Here is cited that the time spent performing a given behaviour may not be enough to assess its welfare relevance, but the variety of behaviour yes.
Is not considered in the paper the richness of behaviours as important tool as welfare analysis, already used in zoos. Discussing behavioural time budgets of zoo-housed individuals based on data collected on the species as well as analysing, the behavioural variety of the subjects can be useful tool that provides some quantitative and qualitative insights into the welfare of individuals and species.
The richness of behaviours is linked to a positive welfare state because a high behavioural diversity indicates that we are meeting several behavioural needs of the animal. On the contrary, when behavioural diversity is low, and animals have no opportunities to perform their behavioural repertoire, they can become lethargic and develop abnormal behaviours. Thus, behavioural variety could be lost during challenging situations that could characterize controlled environments and human management, and the presence of different normal and natural behaviours performed by each subject could indicate a positive welfare state.
Of course all data have to be compare as the autor wrote, with physiological and other data.
references
- Mason, G.J.; Latham, N.R. Can’t stop, won’t stop: Is stereotypy a reliable animal welfare indicator? Anim. Welf. 2004
- Hill, S.P.; Broom, D.M. Measuring zoo animal welfare: Theory and practice. Zoo Biol. 2009
- Spiezio, C.; Valsecchi, V.; Sandri, C.; Regaiolli, B. Investigating individual and social behaviour of the Northern bald ibis(Geronticus eremita): Behavioural variety and welfare. PeerJ 2018,
- Miller, L.J.; Pisacane, C.B.; Vicino, G.A. Relationship between behavioural diversity and faecal glucocorticoid metabolites: A case study with cheetahs (Acinonyx jubatus). Anim. Welf. 2016
- McCormick,W. Recognizing and Assessing PositiveWelfare: Developing Positive Indicators for Use inWelfare Assessment. In Proceedings of the Measuring Behavior, Utrecht, The Netherlands 2012.,].
--> Thank you for the detailed comments and the references. A long mention about Behavioural Diversity Index as a potential welfare indicator has been added in section 3.3.
2 2. Assessment of welfare based on time-budgets.
Is cited the Qualitative Behavioural Assessment’ but could be interesting cite other methodology already used, as the Behavioural Variety Index (BVI)( Spiezio, C.; Valsecchi, V.; Sandri, C.; Regaiolli, B. Investigating individual and social behaviour of the Northern bald ibis(Geronticus eremita): Behavioural variety and welfare. PeerJ 2018) --> A long mention about Behavioural Diversity Index as a potential welfare indicator has been added in section 3.3.
Could be improve the last part: “However, one problem with this methodology is the possibility that preconceived and untested ideas about the needs of animals can bias the welfare ratings provided by keepers. Another possible limitation of making a qualitative assessment of welfare based on the animal’s demeanor, is that it can be more difficult for non-mammalian species”.
In any case Keepers can use observation to understand individuality. (this is another point important for future research)
Know the personality us necessary to plan the space of an enclosure, predict how individuals will react to one another, provide enrichment and apply similar management strategies to reduce eventual frustration and stress. Are important further research combining the personality assessment with trait rating surveys that will further allow comparison between species, and examine how personality impacts welfare. (Tyler C. Andres-Bray, 2020) --> Related to the last three comments/paragraphs. A section discussing the importance of individuality (and temperament) has been added in section 4. (lines 265-272)
- Other points need more detail, as neuroscience approach, personality, enclosure and management: different research found an influence on the behavior, the proximity at the enclosure of specific animals (example Eriksson et al. (2010) found that 30% of zoos in their study situated red panda exhibits adjacent to those of large carnivores. This could lead to chronic stress, which has been linked to poor reproductive and immune functioning (Terio et al. 2004; Mason and Rushen 2006). --> Authors agree with the reviewer in the importance of such elements for the welfare of zoo animals. However, environmental-based indicators (or zoo animal welfare common threats) are not included in this review. A sentence clarifying that environmental-based indicators are not included has been added (lines 256-258)
- Conclusion Please explain in more detail what you mean by ..."the applications to zoo animals of experimental paradigms initially developed for farm and laboratory animals and discussions on the possible pitfalls and methodological and conceptual limitations of some welfare assessment approaches... etc" How can the two scopes be compared? I don't understand --> It refers to the fact that the application of experimental paradigms designed to assess affective states or other welfare-related elements (for instance, the cognitive bias tests) were initially developed for farm or lab animals but have been applied in zoo animals. By doing so, they contributed to the progress in the scientific foundations of zoo animal welfare assessments. And the discussions of “comparative animal welfare” among animal-related areas (lab, farm, free-ranging -wild-, companion, etc.) also contributed to this progress. Few changes have been made in order to facilitate the understanding of this ideas (first paragraph of section 5).